# Competition contributes to both warm and cool range edges

Shengman Lyu [1✉] & Jake M. Alexander [1]

Competition plays an important role in shaping species' spatial distributions. However, it remains unclear where and how competition regulates species' range limits. In a field experiment with plants originating from low and high elevations and conducted across an elevation gradient in the Swiss Alps, we find that both lowland and highland species can better persist in the presence of competition within, rather than beyond, their elevation ranges. These findings suggest that competition helps set both lower and upper elevation range limits of these species. Furthermore, the reduced ability of pairs of lowland or highland species to coexist beyond their range edges is mainly driven by diminishing niche differences; changes in both niche differences and relative fitness differences drive weakening competitive dominance of lowland over highland species with increasing elevation. These results highlight the need to account for competitive interactions and investigate underlying coexistence mechanisms to understand current and future species distributions.

[1] Institute of Integrative Biology, ETH Zürich, 8092 Zürich, Switzerland. ✉email: shengman.lyu@usys.ethz.ch

Climate and biotic interactions, in particular competition, are key factors constraining species' geographic ranges[1–3]. According to niche theory, a focal species is able to overcome abiotic and biotic constraints on population growth to maintain persistent populations within, but not beyond, its range edge[4]. This hypothesis has been widely supported by empirical studies. For instance, transplant experiments have often found declines in fitness when species are transplanted beyond their range edges, with greater declines in fitness when individuals are transplanted into sites where other species are present (see ref. [5] for a review). Another longstanding hypothesis regarding species' range limits is that competition is especially important for setting warm range edges, such as those at low elevation or latitude[1,2,6,7]. However, while some transplant studies have found patterns consistent with this hypothesis[5], other studies indicate that competition is also important for setting cool range edges such as those at high elevation or latitude[8–10]. Thus, although the general importance of competition in shaping distributions is well established, how this varies across environmental gradients, as well as the underlying processes involved, remain unclear.

We can use tools and insights from community ecology theory to better understand when competition contributes to setting range limits[11,12]. From the perspective of coexistence theory, within its range a population of a focal species is predicted to be able to grow from low density when other species are at their equilibrium abundance (i.e. invasion population growth rate, $\ln(\lambda_{\text{invasion}}) > 0$) and thus persist with other community members. When beyond its range limit, the same species is predicted to be unable to maintain persistent populations (i.e. $\ln(\lambda_{\text{invasion}}) < 0$), resulting in competitive exclusion[11], even if the species would be able to persist in the absence of competitors (i.e. intrinsic population growth rate, $\ln(\lambda_{\text{intrinsic}}) > 0$). In other words, the ability of a species to coexist with its neighbours is expected to be reduced towards its range edge.

A key advantage of conceiving range limits as a coexistence problem is that we can begin to unpack the processes through which competitors influence species' range limits[12]. Modern coexistence theory predicts that coexistence is determined by two sets of differences between competing species[13]. Differences in competitive ability (i.e. relative fitness differences) reflect both intrinsic differences in species' demographic performance and/or their sensitivity to competition, and drive competitive exclusion[14]; species' differences that reduce the intensity of interspecific relative to intraspecific competition (i.e. niche differences) promote stable coexistence[15]. These can result for example from resource partitioning or specialist natural enemies[15]. Accordingly, competition could shape range limits through two pathways. Firstly, a focal species may be competitively eliminated outside of its range because it encounters other species with which it has greater niche overlap compared to species within its range. For example, the replacement of closely related, ecologically similar species across elevation gradients has been explained by interspecific competition for shared habitats[16]. Similarly, changing environmental conditions towards range edges can also reduce niche differences between currently co-coexisting species (i.e. decreased niche differences toward the range edge, see ref. [17] for a similar example), increasing the likelihood of competitive exclusion.

The second pathway through which competition could contribute to range limits is via changes to relative fitness differences. A focal species may coexist stably with other species within its range but be competitively excluded beyond its range edge because of reduced competitive ability. On the one hand, reduced competitive ability could arise because the focal species has a reduced ability to tolerate environmental conditions outside of its range in the absence of other competitors (i.e. a diminished $\lambda_{\text{intrinsic}}$). On the other hand, the focal species may become more sensitive to competition, or its competitors exert stronger effects beyond versus within its range[15,18]. Taking an elevation gradient as an example, a high elevation plant could be restricted to higher elevations by competition from lower elevation species that are better competitors for shared resources, like light[19], under a warmer climate[20,21]; but even competition with co-occurring high elevation species might generate a range limit, if at least some of those species become relatively stronger competitors under warmer conditions at lower elevation (see refs. [22,23] for similar examples).

Here, we evaluate the hypothesis that species experience a reduced ability to coexist with neighbours beyond versus within their range edges, and examine the contributions of niche and relative fitness differences to changing coexistence across an elevation gradient. To address this, we conducted a field experiment interacting seven low and seven high elevation species with limited range overlap (hereafter lowland and highland species) in three sites across an elevation gradient. The low (890 m a.s.l.) site was located within the current elevation range of lowland species and beyond the lower range edge of highland species, while the high site (1900 m) was located within the elevation range of highland species and beyond the upper range edge of lowland species, and the middle site (1400 m) was located near the shared range edges of both groups. In each site, we parameterized integral projection models (IPM) to estimate population growth rates ($\lambda$) both in the absence (i.e. intrinsic population growth rates) and presence (i.e. invasion population growth rates) of neighbour species. Our approach enabled us to quantify coexistence determinants (i.e. niche differences and relative fitness differences) and predict the outcomes of competition between species pairs. We ask two main questions: (1) How does the ability of lowland and highland species to persist with competitors change across an elevation gradient? (2) To what extent are any changes in the strength of coexistence driven by changes in niche differences and changes in relative fitness differences? Our study shows that both lowland and highland species display a reduced ability to persist with competitors beyond their range edges, suggesting that competition helps set range limits both at low (i.e. the lower range edges of highland species) and high (i.e. the upper ranges edges of lowland species) elevations. Furthermore, we show that competition can influence species' elevation ranges through changes to both niche differences and relative fitness differences, but that their relative importance varies depending on the origin of the competitors. These results therefore highlight the importance of considering competition and its underlying mechanisms to understand species' distributions across environmental gradients, and ultimately to forecast range dynamics as environments change.

## Results

**Declines in intrinsic and invasion population growth rates towards range edges.** Lowland and highland species displayed distinct responses to elevation in the absence of neighbours (Fig. 1a). The intrinsic population growth rates of both lowland and highland species were projected to decline beyond elevational range edges (test based on the mean of 500 bootstraps; elevation × species origin: $F_{1,41} = 7.062$, $P = 0.008$; Supplementary Table 4; median and 95% CI of elevation × species origin interaction across bootstrap replicates: 0.0011, 0.0008 to 0.0013). Interestingly, most species were projected to be able to persist in the absence of neighbours across the whole elevation gradient, regardless of their elevation origin (i.e. $\ln(\lambda_{\text{intrinsic}}) > 0$; Supplementary Fig. 6).

In the presence of neighbours, the projected population growth rates displayed greater declines beyond the range edges of both lowland and highland species (Fig. 1b; test based on the mean of 500 bootstraps; elevation × species origin: $F_{1,300} = 21.215$,

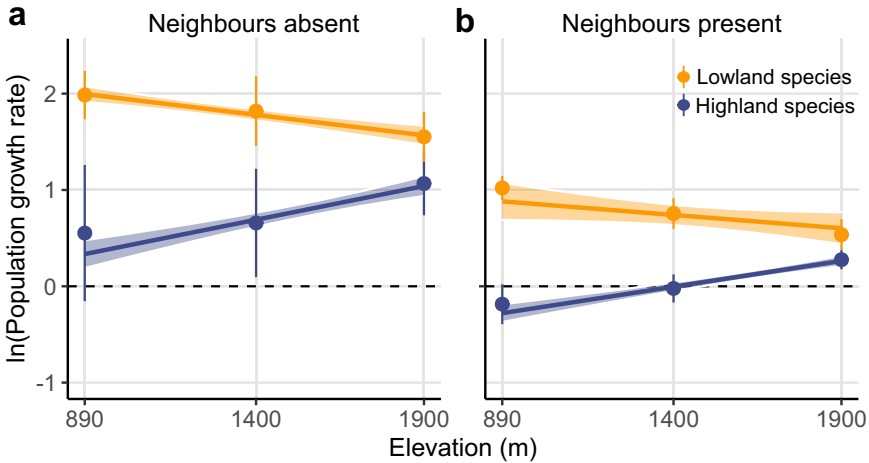

**Fig. 1 Population growth rates across the elevation gradient.** Population growth rates (λ) of lowland (orange) and highland (blue) species across the elevation gradient when neighbours are absent (**a**; n = 7 at each elevation for lowland species, n = 6, 7, and 7 for highland species at the low, middle, and high elevations, respectively) and present (**b**; n = 51, 51, and 56 for lowland species, and n = 40, 48, and 54 for highland species at the low, middle, and high elevations, respectively). Populations are predicted to persist when ln(λ) > 0 (dashed horizontal line). The points and error bars represent the averages and standard errors across species at each site. Solid lines show significant relationships (test based on the mean of 500 bootstrap replicates; P < 0.05; Supplementary Table 4). Shading indicates 95% confidence intervals for the regressions derived from bootstrap replicates of the dataset (N = 500 bootstraps), indicating the propagated uncertainty in the relationships resulting from model parametrization (see Methods). Source data are provided as a Source Data file.

P < 0.0001; Supplementary Table 4; median and 95% CI of elevation × species origin interaction across bootstrap replicates: 0.0008, 0.0005 to 0.0011). When competing against other species, highland species were on average predicted to be able to persist at the high site, that is within their range, but not beyond their lower range edges at the low elevation site. In contrast, the projected ability of lowland species to persist under competition decreased with increasing elevation, though most of them were predicted to be able to persist in all three sites (Fig. 1b; Supplementary Fig. 6).

**Changing outcomes of competition across the elevation gradient.** The predicted ability of species to coexist changed across the elevation gradient, with effects that depended on competitor identity (Fig. 2). For sympatric pairs (i.e. lowland–lowland and highland–highland pairs), coexistence was more prevalent within than outside of the current ranges. On average across all bootstrap replicates, 63% of all eight and 67% of all five highland–highland pairs were predicted to coexist at the high and middle site versus only 40% at the low site (Fig. 2c). Predictions of coexistence for lowland–lowland pairs were more common and displayed only a few changes across sites, with 78% of all 10 lowland–lowland pairs predicted to coexist at the low and high sites and 50% at the middle site (Fig. 2a). Allopatric pairs (i.e. lowland–highland pairs) were more likely to coexist at the middle elevation near shared range edges, with 58% of all 13 pairs predicted to coexist at the middle site and 46% and 44% at the low and high sites, respectively (Fig. 2b).

These findings based on tallies of coexistence outcomes were supported by an analysis of a metric measuring the strength of coexistence (i.e. the coexistence metric, see Methods). These showed that both lowland–lowland and highland–highland pairs displayed weakened coexistence (i.e. decreasing coexistence metric) beyond their elevation ranges, while lowland–highland pairs displayed the greatest ability to coexist at the middle elevation (Fig. 2d-f; test based on the mean of 500 bootstraps: elevation × competitor identity: $F_{2,90} = 11.011$, P = 0.004; Supplementary Table 5). The interaction was mainly driven by the opposing responses of lowland and highland pairs, although this contrast was not significantly different from zero after error

propagation (individual tests for each bootstrap replicate; median and 95% CI: −0.0006, −0.0017 to 0.00004).

**Niche differences and relative fitness differences jointly mediate changing coexistence.** Changes in the magnitude of both niche and relative fitness differences mediated variation in coexistence across the elevation gradient (Fig. 3). For sympatric pairs, both lowland and highland species displayed greater niche differences (i.e. smaller niche overlap) in sites that lay within their range (Fig. 3a, c; test based on the mean of 500 bootstraps; elevation × competitor identity: $F_{2,90} = 11.603$, P = 0.003; Supplementary Table 5), although this result was not significant after error propagation (individual tests for each bootstrap replicate; median and 95% CI of the contrast between lowland and highland pairs: 0.0005, −0.00005 to 0.0012). In addition, reduced relative fitness differences towards high elevation also contributed to the strengthened coexistence of highland species within their range (Fig. 3f; test based on the mean of 500 bootstraps; $F_{1,17} = 5.179$, P = 0.023), although this result was not significant after error propagation (individual tests for each bootstrap replicate; median and 95% CI: −0.0002, −0.0006 to 0.0001). No significant trend in relative fitness differences was seen for lowland–lowland pairs ($F_{1,29} = 0.192$, P = 0.661) or when pooling all pairs together and taking absolute values (i.e. absolute differences in competitive ability irrespective of competitor origin; test based on the mean of 500 bootstraps, $F_{1,90} = 1.406$, P = 0.236; Supplementary Table 5).

For parapatric pairs, lowland species were predicted to be competitively dominant over highland species in general (i.e. ln(fitness difference) < 0), but their competitive dominance was predicted to decrease towards high elevation, where some highland species were predicted to be dominant (Fig. 3e; test based on the mean of 500 bootstraps, $F_{1,44} = 3.379$, P = 0.046). However, this result was not significant after error propagation (individual tests for each bootstrap replicate; median and 95% CI of the slope: 0.0001, −0.0002 to 0.0005). Niche differences between lowland and highland species tended to be greatest at the middle elevation (Fig. 3b), which explained their greatest ability to coexist at the middle site in conjunction with reduced relative fitness differences (Fig. 2e).

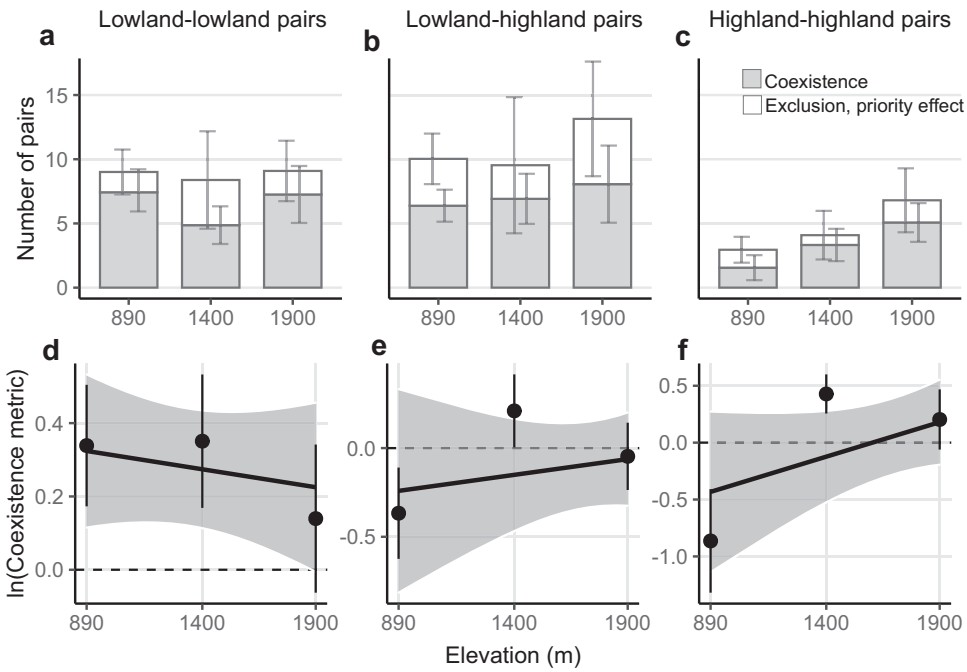

**Fig. 2 Competition outcomes across the elevation gradient.** The estimated competitive outcomes (first row) and strength of coexistence (i.e. coexistence metric, log-transformed; second row) across the elevation gradient of lowland–lowland (**a**, **d**; $n = 9$, 10, and 10 pairs at the low, middle, and high elevations, respectively), lowland–highland (**b**, **e**; $n = 14$, 12, and 18 pairs at the low, middle and high elevations, respectively), and highland–highland species pairs (**c**, **f**; $n = 4$, 5, and 8 pairs at the low, middle and high elevations, respectively). In **a**–**c**, stacked bars represent the mean number of pairs showing stable coexistence (grey) and competitive exclusion or priority effects (white), with error bars showing 95% confidence intervals of each group based on the bootstrap replicates ($N = 500$ bootstraps). In **d**–**f**, the points and error bars represent the averages and standard errors across species pairs at each site; solid lines indicate a significant interaction of elevation × competitor identity (test based on the mean of bootstrap replicates; $p < 0.05$; Supplementary Table 5); shading indicates 95% confidence intervals for the regressions derived from bootstrap replicates of the dataset ($N = 500$ bootstraps), indicating the propagated uncertainty in the relationships resulting from model parametrization (see Methods). Source data are provided as a Source Data file.

## Discussion

Although the role of competition in shaping species' spatial distribution is well acknowledged[2,3], how this role varies across environmental gradients, and by which processes (e.g. through changes in niche and relative fitness differences) competition regulates species' range limits, remains unclear[6,12]. Here we found that the ability of species to persist with competitors was weakened beyond versus within their elevation ranges, supporting the role of competition in shaping species' elevation distribution limits. However, in contrast to the conventional view that competition is more important in abiotically benign environments, such as at low elevation or latitude (e.g. as predicted by the stress-gradient hypothesis[6,7,24]), our results indicate that competition can also be important for high elevation range limits. There are at least two aspects that distinguish our study from previous work that can partly help explain this discrepancy. Firstly, we focused on the population-level outcomes of competition rather than the individual-level strength of competition, as most previous empirical studies have done[5,25]. Both theoretical and empirical work demonstrate that competition can be equally important for population dynamics in harsh and benign environments[10] and its effects cannot be completely understood by focusing solely on the strength of competition[9,26]. For example, harsh environmental conditions could reduce population growth rates directly, making species less tolerant of interspecific competition and competitive exclusion correspondingly more likely to occur, despite any reduced intensity or prevalence of interspecific competition itself[9,27,28]. Secondly, we grew the same set of species in each experimental site, which allowed us to isolate changes in the intensity of competition from turnover in community composition across the environmental gradient[6]. Although this to some extent

divorces our study from the effects of the complex interactions that play out in natural multispecies communities (see ref. [29] and below), it shows that both changing intensity of interactions and changing identity of species contribute to the changing competition experienced by a focal species towards its range edge, and we obtained further insight into these processes through analyses of changes in niche and fitness differences across elevation.

Although effects of niche and relative fitness differences cannot be understood in isolation[13], for our species, changes in niche differences appeared to be especially important for mediating changing coexistence between species across the elevation gradient. Sympatric species (i.e. lowland–lowland and highland–highland pairs) possessed greater niche differences within versus outside of their elevational range, while parapatric species (i.e. lowland–highland pairs) displayed the greatest niche differences at the middle elevation site near to their shared range margins. So far, there have been few a priori predictions made about how niche differences should change across environmental gradients. Nonetheless, our finding of greater niche differences between sympatric species within than beyond their range is consistent with a recent study showing reduced niche differences under drought conditions because drought increased the intensity of interspecific but not intraspecific competition[30]. We can only speculate on the biological processes driving the distinct responses of niche differences without further information (e.g. from functional traits[31]). However, one possible explanation could be temporal overlap in phenology. Specifically, the flowering phenology of lowland and highland species overlapped most at the high and low elevation sites, respectively, while the overlap between lowland and highland species was smallest at the middle elevation site (Supplementary Fig. 7). Thus, the phenological overlap between species was associated with the magnitude of niche differences, with

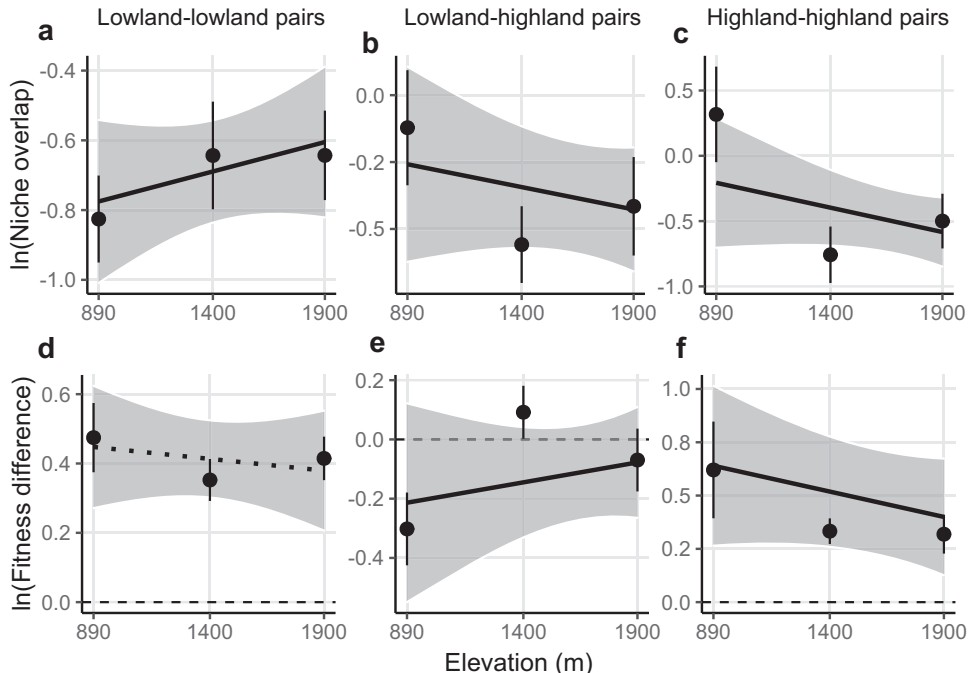

**Fig. 3 Niche differences and relative fitness differences across the elevation gradient.** Estimates of the niche differences (expressed as niche overlap, i.e., 1—niche differences; see Methods; first row) and relative fitness differences (second row) across the elevation gradient of lowland–lowland (**a**, **d**; $n = 9$, 10, and 10 pairs at the low, middle, and high elevations, respectively), lowland–highland (**b**, **e**; $n = 14$, 12, and 18 pairs at the low, middle, and high elevations, respectively) and highland–highland species pairs (**c**, **f**; $n = 4$, 5, and 8 pairs at the low, middle, and high elevations, respectively). Both niche overlap and relative fitness differences are shown on a log scale. The points and error bars represent the averages and standard errors across species pairs at each site. Solid lines indicate a significant interaction of elevation × competitor identity for niche differences, and significant effects of elevation on relative fitness differences for each type of pair separately (test based on the mean of bootstrap replicates; $p < 0.05$); the dotted line indicates a non-significant relationship ($p > 0.05$; Supplementary Table 5). Shading indicates 95% confidence intervals for the regressions derived from bootstrap replicates of the dataset ($N = 500$ bootstraps), indicating the propagated uncertainty in the relationships resulting from model parametrization (see Methods). In **d**–**f**, absolute values of fitness differences are shown for lowland–lowland and highland–highland pairs, while actual values are shown for lowland–highland pairs, with negative values indicating lowland species are dominant competitors. Source data are provided as a Source Data file.

greater phenology overlap leading to smaller niche differences and vice versa[32,33].

Lowland species were in general competitively superior to highland species, but this superiority decreased significantly with increasing elevation, and the competitive dominance of some species pairs was switched in favour of highland plants at the high elevation site (i.e. ln(fitness difference) > 0 in Fig. 3e). Nonetheless, in most cases the competitive advantage of highland species at the high site was not great enough to exclude the lowland species; possible reasons for this are discussed below. The competitive dominance of lowland species is consistent with general expectations[6] (but see ref. [20]), as well as empirical evidence showing that lowland species can exert stronger competitive effects on alpine focal plants than other alpine neighbours[21]. Greater competitive superiority of species within their home range could result either from their higher intrinsic population growth, in particular for highland species, or from the fact that species are more resistant to competition within versus beyond their range[18,32], although we cannot isolate the relative contribution of these two processes with the data at hand. Similarly, the increased relative fitness differences of highland species towards low elevation could result from amplified differences in intrinsic population growth or competitive responses[30]. Taken together, our results demonstrate that the range limits of our focal species were influenced by competitors from both within and outside of their range (i.e. sympatric and parapatric species) and mediated through both niche differences and relative fitness differences. These results speak to the need to broadly investigate how environmental conditions impact niche and relative fitness

differences and how they jointly structure communities and species' distributions across environmental gradients[12].

Although we found significant effects of elevation on the strength of coexistence (i.e. the coexistence metric), we observed relatively few cases in which coexistence outcomes were qualitatively altered across the elevation gradient. For example, most lowland species could still maintain relatively large and positive invasion growth rates, and thus were predicted to persist, at the high site. The high site is above the current range limit of all lowland species in this region, and so it is surprising that the climatic and biotic environment appears to be suitable for them. This might be because while average climate conditions at high elevation support the growth of our lowland plants, climate extremes do not[34]; if sufficiently extreme conditions did not occur during the experiment, then climate limitation would not be observed. Another possible explanation for lowland species' persistence at high elevation is disequilibrium between their distributions and current climate. It has been widely reported that the rate of species' upwards migration across elevation gradients lags behind the pace of climatic warming[35–37]. For example, the observed average upward shifts of 183 plant species over the past 50 years in the European Alps was 30 m[38], while isoclines in the same region have shifted upwards by 310–355 m[35]. Disequilibrium in the distributions of our lowland species might be due to their limited dispersal capacities and the often low rate of germination and establishment of perennial species[36]. Added to this, the growing season temperatures over the course of our experiment generally exceeded the long-term average (https://www.meteoswiss.admin.ch/). Thus, the high site could already be

within the potential upper limits of some of the lowland species included in our study. But even if this were the case, the relationship shown in Fig. 1 implies that lowland species ranges will be limited by competitors before they are limited by climate.

Manipulative experiments such as ours will tend to underestimate the prevalence of stable coexistence when they focus on pairwise competition and use homogeneous environmental conditions over a relatively short temporal scale (e.g. ref. [14]). Firstly, such experiments might miss potentially important stabilizing effects that only emerge from interaction networks in which any given pair of species is not necessarily able to coexist (e.g. higher-order interactions[29,39]). Secondly, standardized, short term experiments might not be able to capture temporal and spatial variation that could be important for species coexistence (e.g. temporal and spatial storage effects[40]). In our experiment, we used the same soil in the three sites to isolate the effects of changing climate from changing soil conditions across the elevation gradient, which might have reduced opportunities for coexistence driven by interactions belowground (e.g, plant-soil feedbacks[41]). This failure to capture all possible coexistence mechanisms might partly explain the lack of stable coexistence between currently co-occurring species observed in our experiment. For example, we found that only 63% of highland species pairs were predicted to be able to coexist at the high site; small-scale environmental heterogeneity has been suggested to be especially important for maintaining the high diversity of alpine plants[42]. Thirdly, due to the short duration of the experiment (four years), our results might be contingent on specific weather conditions over the course of the experiment, which may also explain the lack of coexistence of highland species within their range. Lastly, violations of the assumption that monocultures are at equilibrium abundance could affect our predictions of coexistence. In our experiment, 32% (11 of 34) of monocultures were predicted to meet this assumption, but those that did not (10 and 13 monocultures were predicted to be above and below equilibrium, respectively) were evenly distributed across the sites and so are unlikely to affect the general patterns we observed (Supplementary Fig. 5).

Our results challenge the conventional view predicting a diminished role for competition in structuring species' distributions and ecological communities under "harsh" environments at high elevation or latitude, whilst highlighting the importance of considering species' interactions to understand and project future species distributions[27,28]. More generally, our finding that both lowland and highland species experienced weakened coexistence towards their range edges (i.e. the upper ranges of lowland species and the lower ranges of highland species, respectively) suggests that competition might be important for setting both species' lower and upper elevation limits[10]. An important role for competition at high elevations could help explain the limited ability of some species to shift their distribution limits to a higher elevation to track climate warming[35,36]. However, our results suggest that continuous warming may facilitate the upwards shifts of lowland species and the local extinction of alpine species[43] by removing the competitive advantage that alpine species possess under a cooler climate. In addition, our results suggest that climatic warming may destabilize alpine plant communities by reducing niche differences and amplifying fitness differences between alpine species.

Although modern coexistence theory provides a useful framework for understanding and predicting species coexistence across environmental gradients, niche differences and relative fitness differences are only phenomenological syntheses of biological processes underlying species interactions. Future work should seek to uncover the mechanisms through which competition influences species' distributions along environmental

gradients by manipulating specific abiotic and biotic factors and gathering data on species' physiology, morphology or life-history strategies (i.e. functional traits[31]). Future studies could also look at whether competition sets range limits across entire species ranges, including upper and lower edges of the same species, and how complex interaction networks in multispecies communities contribute to setting range limits. Such studies would help to provide more mechanistic insights into the role of species interactions in regulating species' distributions and bolster our ability to more accurately forecast the impacts of environmental change on species' distributions and community dynamics.

## Methods

**Study area and species**. We selected three sites across an elevation gradient in the western Swiss Alps (Bex, Canton de Vaud), situated at 890, 1400 and 1900 m above sea level (hereafter, the low, middle, and high sites; Supplementary Fig. 1). The three sites span a temperature gradient ranging from 2.5 to 9.6 °C (mean annual temperature from 1981 to 2015[44]; Supplementary Table 1). With increasing elevation, soil moisture increased, and the growing season length was shortened by a longer snow-covered period, as measured from July 2019 to June 2020 (Supplementary Fig. 2). All sites were established on south-facing and shallow slopes in pasture and fenced to exclude livestock.

We included 14 herbaceous focal species that frequently occur in this region, half of which originated from low elevation (hereafter lowland species) and half from high elevation (highland species, Supplementary Table 2). Lowland species had upper range limits (defined as the 90th percentile of their elevation distribution) below 1500 m (with the exception of *Plantago lanceolata*, with a 90th percentile of 1657 m), while highland species had lower range limits (defined as the 10th percentile of their elevation distribution) above 1500 m, based on a dataset of 550 vegetation plots from the study area[45]. These species consisted of 12 perennial and two biennial species, which are the dominant life histories in this region. Species were selected to include a range of functional types (7 forbs, 4 grasses, 3 legumes) and functional traits (based on plant height, specific leaf area and seed mass). Seeds were obtained from regional suppliers given the large quantities that were needed to establish the experiment (Supplementary Table 2).

**Field competition experiment**. We designed a field experiment to study the effects of elevation on population growth rates and competitive outcomes by growing focal plants either without competition or competing with a background monoculture of the same or another species (Supplementary Fig. 1). In spring 2017, we established 18 plots (1.6 × 1 m, 0.2 m deep) at each of the three field sites, lined with wire mesh to exclude rodents (except at the high site) and with weed-suppressing fabric on the sides to prevent roots growing in from outside. To control for soil effects, the beds were then filled with a silt loam soil that originated from a nutrient-poor meadow at 1000 m a.s.l. within the study area. Four plots were maintained as bare soil plots (non-competition plots). The other 14 plots received 9 g m$^{-2}$ of viable seeds of each species, which allowed the establishment of a monoculture of relatively high density (competition plots). We then periodically weeded the plots to maintain monocultures over the course of the experiment. All species except for two (*Arnica montana* and *Daucus carota*) successfully established monocultures, of which 11 species (including six lowland species and five highland species) were fully established by autumn 2017. We then resowed the other plots that failed to establish, which subsequently established either in spring 2018 (*Poa trivialis* and *Poa alpina* in the low site and *Bromus erectus* in the middle site) or autumn 2018 (*Aster alpinus*, *P. trivialis* and *P. alpina* in the middle site and *Sesleria caerulea* in the low and high sites). Species that failed to establish were included only as focal species for the calculation of invasion population growth rates (i.e. the density was low for *A. montana* and *D. carota* in all sites, *Trifolium badium* in the low site and *S. caerulea* in the middle site, probably due to high mortality rates caused by drought).

We first raised focal seedlings of each species in a greenhouse for six weeks on standard compost and then transplanted them into the field (Supplementary Fig. 1). To test for responses to elevation in the absence of competition, focal plants were transplanted into non-competition plots at 25 cm apart in autumn 2017 (n = 9 per species and site). To test for effects of competition, we transplanted focal individuals into established plots with 14 cm spacing (n = 9 per focal species, competitor and site). Focal plants that died within two weeks of transplanting were replaced (ca. 5%), assuming mortality was caused by transplant shock. Note that we transplanted focal plants into plots only when the background monocultures were fully established. In 2018 and 2019, we replaced dead focal individuals in spring and autumn (ca. 10% each time). The full design included 56 unique interspecific pairs in each site accounting for 61% of all 14 × 13 = 91 possible pairwise combinations. These pairs were selected to evenly sample differences in functional trait space based on a pilot analysis using plant height, specific leaf area and seed mass obtained from the LEDA dataset[46]. Each focal species competed against four lowland and four highland species, yielding 14 lowland–lowland and highland–highland pairs and 28 lowland–highland pairs. Across all three sites, this

design resulted in $N = 3780$ individuals in total ([56 interspecific pairs × 2 + 14 intraspecific pairs + 14 non-competition] × 9 individuals × 3 sites).

**Demographic data**. We followed each focal individual between 2017 and 2020 to monitor individual-based demographic performance (i.e. vital rates; Supplementary Fig. 4). Survival was monitored twice a year at the beginning and the end of the growing season. Towards the end of the growing season each year (August–September), we measured all individuals to record plant size, whether they flowered, and to estimate seed production on flowering individuals. To estimate focal plant size, we measured size-related morphological traits on all focal individuals at each census (i.e. the number and/or length of flowering stalks, leaves or ramets, depending on the species) and estimated dry aboveground biomass using regression models fitted using collected plant samples (mean $R^2 = 0.871$; Supplementary Data 1; Supplementary Methods). To estimate seed production, we counted the number and measured the size of fruits on reproductive individuals; we then estimated the number of seeds produced by each individual using regression models fitted using intact fruits of each species collected at the early fruiting stage on background plants (mean $R^2 = 0.806$; Supplementary Data 2; Supplementary Methods). We conducted a separate experiment to estimate the germination and recruitment of each species in each site (Supplementary Methods).

**Population modelling**. To estimate population growth rates ($\lambda$), we built integral projection models to incorporate multiple vital rates across the life cycle[47] (see Supplementary Table 3 for model structure and parameters). Separate IPMs were built to estimate intrinsic growth rates using plants growing in the absence of competition (in non-competition plots) and invasion growth rates using plants growing within the background monocultures (in competition plots), under the assumption that monocultures were at equilibrium (see Supplementary Fig. 5 for a test of this assumption) and that focal individuals did not interact with each other but only with the background species. We used plant size (i.e. estimated dry aboveground biomass, log scale) as a continuous state variable and fitted linear models to estimate vital rate parameters by combining multiple-year demographic data over three censuses (i.e. 2017–2018, 2018–2019, and 2019–2020; see Supplementary Methods for consideration of more complex models). We modelled the probability of survival, flowering, and seedling establishment using generalized linear models with a binomial error distribution, modelled growth and seed production using general linear models and described the offspring size distribution using Gaussian probability density functions. We modelled seed germination, seedling establishment and the seedling size distribution as size-independent functions, assuming they are unaffected by maternal size (Supplementary Fig. 4; Supplementary Table 3). For each vital rate of each species, we selected the best-fitted vital rate model by comparing all nested models of the full models using the Akaike information criterion corrected for small samples (AICc), which allowed us to avoid overfitting models and to borrow strength across competitor species and sites in cases where full models were outperformed by reduced models (Supplementary Methods; Supplementary Data 3 and 4).

We calculated population growth rates ($\lambda$) as the dominant eigenvalue of the IPMs, which represents the discrete per-capita growth rate (i.e. $N_{t+1} = \lambda N_t$)[47]. To evaluate the uncertainty around $\lambda$, we performed parametric bootstraps for size-dependent vital rates (i.e. survival, growth, flowering, and fecundity). Specifically, we resampled the parameters of each vital rate model using multivariate normal distributions based on their means and covariance matrices[48]. We then fitted all IPMs and estimated $\lambda$ for each of the 500 bootstrap replicates (Supplementary Data 5).

**Estimation of niche differences, relative fitness differences, and coexistence outcomes**. We quantified niche and relative fitness differences and predicted coexistence outcomes following the method of Carroll et al.[49]. This method is based on species' sensitivity to competition defined as the proportional reduction of the population growth rate of a focal species $i$ when invading a population of a competitor species $j$ that is at its single-species equilibrium, and is mathematically equivalent to one minus the response ratio:

$$S_{ij} = 1 - \frac{\ln(\lambda_{ij})}{\ln(\lambda_i)} \quad (1)$$

where $\lambda_{ij}$ denotes the invasion growth rate of focal species $i$ and $\lambda_i$ is its intrinsic growth rate. The natural logarithm of discrete population growth rates $\lambda$ estimated from IPMs are equivalent to per-capita growth rate in continuous population growth models[50], and this transformation makes sensitivities compatible with the coexistence analysis described below. Sensitivity is greater than 0 for antagonistic interactions, with higher values equating to stronger competition, while facilitative interactions lead to negative sensitivities.

For a pair of species, modern coexistence theory predicts that niche differences (ND) promote coexistence by reducing the intensity of interspecific competition experienced by both species. Therefore, a pair of species with a large niche difference should display small mean sensitivities to competition from each other. Consequently, niche differences can be calculated as one minus the geometric mean of the two sensitivities (i.e. niche overlap). In contrast, relative fitness differences (RFD) quantify the degree of asymmetry in species' competitive abilities. Therefore, a pair of species with a large fitness difference should display

large differences in their sensitivities to competition from each other, as quantified as the geometric standard deviation of sensitivities[49]:

$$ND = 1 - \sqrt{S_{ij} S_{ji}} \quad (2)$$

$$RFD = \sqrt{S_{ji} / S_{ij}} \quad (3)$$

There are three possible outcomes of competition between a given pair of species: stable coexistence, a priority effect, and competitive exclusion. These can be quantified based on either invasion criteria or the relative magnitude of niche differences versus relative fitness differences[15,51]. Stable coexistence is only possible when both species are able to invade each other's equilibrium populations; this condition is met when $ND > 0$ and $RFD < \frac{1}{1-ND}$[49], which is equivalent to $\frac{1}{RFD(1-ND)} > 1$, with greater values indicating more stable coexistence and providing a metric for the strength of coexistence (i.e., coexistence metric[26]). When neither species can invade when rare, then priority effects occur, meaning that whichever species is initially established within a community has an advantage and excludes the other. This could happen when a species pair has a small niche difference and a small relative fitness difference, that is $ND < 0$ and $RFD < \frac{1}{1-ND}$[51]. Otherwise, if only one species can invade, then it is predicted to competitively exclude the inferior competitor; this occurs when relative fitness differences are big enough to override the stabilizing effect of niche differences, that is when $RFD > \frac{1}{1-ND}$. We quantified competitive outcomes and coexistence metrics for each of the 500 bootstrap replicates of the dataset (Supplementary Data 6).

Note that we excluded facilitative interactions that were present in 13% of all pairs because the equations for niche differences and relative fitness differences are not compatible with negative values of sensitivity (Eq. 2 and 3); we did not exclude facilitative interactions for other analyses. We quantified the coexistence determinants of species pairs in cases where either one or both of the species were predicted to be unable to persist in the absence of neighbours (i.e. $\ln(\lambda_{\text{intrinsic}}) < 0$; 8% of all pairs). This was because the quantification of niche and fitness differences were based on the competitive impacts on population growth (i.e. sensitivity to competition) and independent of the actual estimates of population growth rates.

**Statistical analysis**. We tested the effect of elevation on species' intrinsic and invasion growth rates, and whether these effects differed between lowland and highland species, by fitting mixed-effects models with elevation (continuous), species origin (i.e. lowland or highland) and their interaction as fixed effects and focal species identity as a random effect. To test whether coexistence was strengthened at high versus low elevation, we fitted a mixed-effects model of the coexistence metric as explained by elevation and competitor identity (i.e. the three types of pairs: lowland–lowland, lowland–highland, and highland–highland pairs), with species pair as a random effect. We fitted similar models for niche differences and relative fitness differences (absolute values) to gain insights into whether changes in coexistence were driven predominantly by variation in niche or relative fitness differences. Additionally, we tested the effect of elevation on relative fitness differences for each type of pair separately and used the actual values of lowland–highland pairs since we were interested not only in the magnitude of relative fitness differences but also in the direction of effects (i.e. whether lowland or highland species were dominant competitors). The response variables in these models were log-transformed to meet model assumptions. Note that we therefore had to use niche overlap (1-ND) for the log-transformation instead of niche differences, because niche differences included negative values (Eq. 2). Firstly, we performed these analyses based on the mean of the 500 bootstrap replicates for a given focal species/ background competitor combination and used likelihood ratio tests to determine the significance of fixed effects. To account for the uncertainty around estimated $\lambda$, we also performed individual tests for each bootstrap replicate of the dataset and determined the significance of fixed terms based on whether the 95% confidence interval of estimated coefficients, or contrasts of interest, included zero. All population modelling and statistical analysis were conducted in R version 4.0.3[52].

**Reporting summary**. Further information on research design is available in the Nature Research Reporting Summary linked to this article.

# Data availability
The data we used to estimate species elevation ranges are based on a published study[45] and could be obtained from the author on request. The functional trait data used for species selection are obtained from the LEDA dataset[46] and available at https://uol.de/en/landeco/research/leda/data-files. The data presented in this study are available in Figshare[53], at https://doi.org/10.6084/m9.figshare.19108127.v1. Source data are provided with this paper.

# Code availability
All R scripts used for modelling and statistical analyses are available on GitHub: https://github.com/ShengmanLyu/competition-contributes-to-range-limits and also deposited in Zenodo: https://doi.org/10.5281/zenodo.6395222.

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

## Acknowledgements

We thank Loïc Liberati, Tim Murray, Jessica Joaquim, Cuicui Zhou, Yan Hess, Kai-Hsiu Chen, Megan Stamp, and other members of the Plant Ecology group at ETH Zürich for help with field work. We thank the Commune de Bex and Jean-Louis Putallaz for access to the field sites. We also thank Antoine Guisan and the ECOSPAT group at the University of Lausanne for providing plant community and climatic data. We thank Jonathan Levine and the Plant Ecology group for their feedback on an earlier version of the manuscript. S.L. acknowledges the Chinese Scholarship Council for financial support (No.201706100184). J.M.A. received funding from the European Union's Horizon 2020 research and innovation programme under grant agreement No. 678841.

## Author contributions

J.M.A. designed the experiment. S.L. and J.M.A. collected data. S.L. conducted the modelling and data analysis. S.L. wrote the manuscript with input from J.M.A.

## Competing interests

The authors declare no competing interests.
