## [Peer Review File · Nature Communications]

REVIEWER COMMENTS

Reviewer #1 (Remarks to the Author):

This manuscript uses an ambitious experimental design to test how coexistence mechanisms change across an elevation gradient for species originating from low or high elevation environments. This involves a series of pairwise competition assays, in which a focal species is grown in a background of itself or a single competitor. This basic design is replicated across all possible species pairs and then repeated in three sites of differing elevation. Demographic models were used to project population growth for each species in each treatment and calculate the contributions of niche and fitness differences to coexistence outcomes. The major findings include that (1) most species can invade (i.e., population growth is positive) most sites in the absence of competition; (2) this remains true for lowland species experiencing competition, but most highland species can no longer invade lowland environments in competitive treatments; and (3) species become relatively more sensitive to competition at elevations outside of their elevation range, due to idiosyncratic contributions of changing niche and fitness differences. I found this manuscript extremely easy to read and understand and the results are quite interesting. I have only a few minor queries and comments.

(1) Please better justify why drop only the two highland species from the ANOVA instead of others with similar issues, e.g. *Poa*.

(2) L139-140: Is this entirely attributable to competition, or were these pairs including the species that could not establish on their own?

(3) Weather during experiment was warm relative to historical normals, so high-high pairs were being tested to coexist under atypical (more lowland-like) conditions. L259-60 attributes their difficulty coexisting to soil, but the prior paragraph about climate change seems at least as likely as a mechanism. Please revise the discussion to include this point.

Reviewer #2 (Remarks to the Author):

This paper asks whether the strength of pairwise competition varies across elevational ranges of alpine species, and tries to identify the mechanism(s) driving differences in competition among and between high- v. low- elevation community members. The authors conduct manipulative experiments of competition to generate estimates of low- & high-density population growth rate, as well as population growth rate when grown with a competitor monoculture, for an impressive array of species spanning a dramatic elevational range. The authors use these results to quantify the strength of competitive effects within v. across species' ranges, as well as to parameterize a statistical method that disentangles the role of niche differences v. relative fitness differences in mediating competition. This manuscript synthesizes an impressive array of field and modelling work, for multiple species, and is novel in that it both quantifies competition outcomes and tries to identify the mechanisms driving competition

outcomes—and how those mechanisms vary geographically. I have a few major points and some minor comments.

Major points:

1. The authors quantify competition between species pairs in monocultures. I understand that these monoculture competition experiments are necessary to parameterize the Carroll et al. approach they use to quantify mechanisms of competition. But these experimental manipulations are quite divorced from how competition actually works in a diverse community. I have a hard time understanding whether the results in Fig 1, which show that the effect of competition (with monocultures) varies systemically with elevation, actually reflect the impact of competition in nature, where species compete not with a monoculture but with diverse communities. This problem seems particularly concerning if there are differences in diversity across elevations (which I imagine there are). The fact only some species pairs that currently co-exist in nature were expected to coexist according to the modelling results reflects this discrepancy.
2. Perhaps the most interesting result in this paper is that niche differences tend to modulate coexistence among sympatric but not parapatric species. However, I found this result somewhat unsatisfying in that the biological mechanism driving this result is not at all clear. While the authors do identify one mechanism that might be driving this effect (phenology), it is not clear to me from their explanation whether such a mechanism might be universal or not, particularly since they provide no biological/ physiological explanation for why phenology overlaps more outside of a species' range rather than within the range.
3. Somewhat related to no. 2, I think the explanation of the Carroll et al. approach needs quite a bit more explanation earlier in the paper, ideally with a biological example to illustrate what these values actually reflect—perhaps even with a conceptual figure of some kind. It is difficult to intuit what the ND v. RFD values actually mean biologically, even after carefully reading the methods a number of times and looking at the Carroll paper. I think this problem is exacerbated by the switching back and forth between “niche difference” v. “niche overlap” in the text.

Minor comments:

Line 14-15: I found this sentence much too vague. I suggest inserting “the importance of” after “how”. I also found “processes” too vague.

Line 48: change “species interactions” to “competition”, as this paper is only focusing on competition, not on all species interactions

Line 124: Missing “growth rates” after ‘intrinsic population.’

The fact that lowland species are predicted to occur at the highland site (but currently do not) is troubling. It is unclear if this phenomenon arises from disconnect between modelling results and reality, or whether it is actually due to disequilibrium between current climate and the species' distributions.

The authors' claim that it is the latter would be strengthened by some evidence of dispersal limitation.

What is the AIC (or AICc?) weight of the top models?

It appears that the issue of size eviction from the IPMs was assessed visually somehow? I'm not sure how you would do this—a more methodological approach (i.e., testing for any eviction) would be more appropriate.

I do not understand the rationale behind randomly selecting 56 IPMs in the supplemental information. More clarification is needed.

In the methods the authors indicate that the monoculture gardens weren't fully established until autumn 2017, but then they use data from the 2017-2018 transition to parameterize their IPMs—so this transition began before the monocultures were fully established and should be excluded.

Reviewer #3 (Remarks to the Author):

This paper combines field experiments with model simulations to look at species range limits using modern coexistence theory. I like the design of this study that directly tests the effects of intra- and interspecific competition and abiotic factors on species coexistence along the elevation. I am impressed by the detailed fieldwork conducted by the authors. However, I have a few suggestions regarding the authors' analysis and presentation of results.

1. As the authors correctly understand, the resident populations of competing species must be at equilibrium when conducting invasion analyses. That is, the population growth rate of the competing species in interspecific competition treatment should be close to zero. However, I did not see the authors address this issue. From Figure S5, it appears that the population growth rates of intra-specific competition treatment are much greater or less than zero in many species. This means that populations are either declining or increasing. Is this also true for resident species in the inter-specific competition treatment? Furthermore, although it is reasonable that the population growth rate for the intra-specific competition is lower than that for interspecific competition (because intra-species competition is higher than the interspecies competition), the population growth of focal species is usually very high, much higher than that of intra-species competition treatment, as shown in Figure S5. This part is very strange to me, do the authors have any explanation? This also leads me to ask whether the competing populations have not reached equilibrium. I hope this part can be discussed more clearly because it is a key assumption of the modern coexistence theory.

2. This also makes me want to see more original data points in the authors' results. The authors provide the formula used to estimate each fitness component in Table S5, as well as the AIC values for the model described in the subsequent table. However, in the absence of comparisons, the AIC values do not provide much information. Therefore, I would like to see direct plots with actual data points of the relationship between the data points and the regressions. I would also like to see the regression plots with and without outliers (authors said they remove less than 1% of the data points) and the effect of these outliers on the results.

3. More formally, as the authors cite in Ellner's (2016) book on IPM estimation (p. 30), Ellner suggests that "Always quantify your uncertainty!", I would like to see the authors estimate the uncertainty of the results and then separate the conclusions for species with higher uncertainty from those with higher statistical power.

4. I read the author's description of the experimental methods many times before I understood how the authors' setups of the intra- and interspecific competition treatment. I suggest that the authors could draw a diagram so that readers can understand how intraspecific competition is created and how the

positions of the resident and invasive populations are arranged in the interspecific competition treatments.

Reviewer #4 (Remarks to the Author):

It was a delight to read this manuscript that uses decent macroecological experiments to examine how biotic interactions shape species distribution boundaries. Such studies are relatively rare, but they are crucial in clarifying the role of complex species interactions. Particularly, the authors applied coexistence theory to understand the processes of competition underlying boundary formation. I had many questions at the beginning of reading this manuscript, but most of these were resolved as I continued. Overall, this is an important study with rigorous experiments, and the manuscript is well-structured. I have a few comments for the authors.

Please forgive me for being a bit picky about the concluding sentences in the abstract. "Taken together, these results challenge the view that competition has a diminished role in structuring communities in abiotically stressful environments." This sentence insinuates a comparison between competition and climate factors, but testing their relative importance was outside the main focus of this study. The experiments were more about 'how' competition processes affected distribution across environmental gradients. The setting of experiments and analytical strategy will inevitably underestimate the effects of abiotic factors, including the use of the same soil substrate, the short time frame for parameter collection, which does not truly reflect how environmental fluctuation affects population growth, and largely focused on niche difference versus fitness differences. In any case, environmental effects are inferential and relative contributions to competition are difficult to present. The authors are well aware of these limitations in the discussion. I suggest that the abstract could be modified to explicitly reflect the focus of the study.

I also suggest that the Introduction paragraph can be presented in a hypothesis-prediction format. The application of coexistence theory in biogeographic studies at a large scale and range limits is still in its infancy (e.g. Alexander et al. 2018 TREE). Its novelty and importance are noted, but not necessary to readers outside the field. The authors could provide more context to smoothly bridge these fields. Particularly in the context of environmental gradients, the so-called harsh or benign climate, what are the expected consequences for niche difference and relative fitness, based on known hypotheses.

The population models are essential to derive the coexistence metrics, niche differences, and relative fitness differences. However, because the parameters were obtained at a short time period, which does not reflect how environmental fluctuation affects species coexistence, the metrics may be systematically biased, probably leading to the underestimation of species coexistence at high altitudes. This may be beyond the scope of this study, but the authors can discuss the possible effects.

The results showed that neither competition nor climate limits lowland species, and their current distribution reflect disequilibrium with climates. The argument will then suggest that the current experimental setting may be somewhat biased and the results can be misleading to a certain extent.

The authors use a community perspective to analyze how the upper and lower boundaries of species are affected by competition, but the comparisons are between the upper boundaries of lowland species and the lower boundaries of highland species, which are all adjacent to other species, not at the harsh boundary of the target gradient, and not comparing the upper and lower boundaries of specific species. As the stress gradient hypothesis can be largely from a species perspective, discussing the relative pressure at their upper and lower boundaries, it is suggested that the authors clarify the differences.

**Author responses**

Reviewer #1 (Remarks to the Author):

This manuscript uses an ambitious experimental design to test how coexistence
mechanisms change across an elevation gradient for species originating from low or
high elevation environments. This involves a series of pairwise competition assays,
in which a focal species is grown in a background of itself or a single competitor.

This basic design is replicated across all possible species pairs and then repeated in
three sites of differing elevation. Demographic models were used to project
population growth for each species in each treatment and calculate the contributions
of niche and fitness differences to coexistence outcomes. The major findings include
that (1) most species can invade (i.e., population growth is positive) most sites in the
absence of competition; (2) this remains true for lowland species experiencing
competition, but most highland species can no longer invade lowland environments
in competitive treatments; and (3) species become relatively more sensitive to
competition at elevations outside of their elevation range, due to idiosyncratic
contributions of changing niche and fitness differences. I found this manuscript
extremely easy to read and understand and the results are quite interesting. I have
only a few minor queries and comments.

Thank you for your suggestions on the analysis and phrasing that helped us to
reduce possible confusion and refine the discussion. We are delighted that the
reviewer found our manuscript easy to understand and interesting.

(1) Please better justify why drop only the two highland species from the ANOVA
instead of others with similar issues, e.g. *Poa*.

We no longer exclude these species in our revision since the sensitivity of this result
can now be fully evaluated using the bootstrapping procedure that we implemented
in response to the comments of reviewer 3.

(2) L139-140: Is this entirely attributable to competition, or were these pairs including
the species that could not establish on their own?

The species that failed to establish as background species were predicted not to
persist in the absence of competition by our models, suggesting the failure to
establish may not be due to competition but due to the abiotic environment, such as
drought at the lowest site. We have added this information to the revised manuscript
(lines 364 - 367) and excluded the original statement that appeared unclear and no
longer relevant since we updated the result on competition outcomes (Fig. 2a-c).

(3) Weather during experiment was warm relative to historical normals, so high-high
pairs were being tested to coexist under atypical (more lowland-like) conditions.
L259-60 attributes their difficulty coexisting to soil, but the prior paragraph about
climate change seems at least as likely as a mechanism. Please revise the
discussion to include this point.

This is a good point. We have included it in the revised discussion (lines 287-290).

Reviewer #2 (Remarks to the Author):

This paper asks whether the strength of pairwise competition varies across
elevational ranges of alpine species, and tries to identify the mechanism(s) driving
differences in competition among and between high- v. low- elevation community
members. The authors conduct manipulative experiments of competition to generate
estimates of low- & high-density population growth rate, as well as population growth
rate when grown with a competitor monoculture, for an impressive array of species
spanning a dramatic elevational range. The authors use these results to quantify the
strength of competitive effects within v. across species' ranges, as well as to
parameterize a statistical method that disentangles the role of niche differences v.
relative fitness differences in mediating competition. This manuscript synthesizes an
impressive array of field and modelling work, for multiple species, and is novel in that
it both quantifies competition outcomes and tries to identify the mechanisms driving
competition outcomes—and how those mechanisms vary geographically. I have a
few major points and some minor comments.

Thank you. We are pleased that the reviewer considers our work to be interesting
and novel.

Major points:

1. The authors quantify competition between species pairs in monocultures. I
understand that these monoculture competition experiments are necessary to
parameterize the Carroll et al. approach they use to quantify mechanisms of
competition. But these experimental manipulations are quite divorced from how
competition actually works in a diverse community. I have a hard time understanding
whether the results in Fig 1, which show that the effect of competition (with
monocultures) varies systemically with elevation, actually reflect the impact of
competition in nature, where species compete not with a monoculture but with
diverse communities. This problem seems particularly concerning if there are
differences in diversity across elevations (which I imagine there are). The fact only
some species pairs that currently co-exist in nature were expected to coexist
according to the modelling results reflects this discrepancy.

We agree with the reviewer that effects of competition in natural multispecies
systems might be more complex, for example because of interaction chains and
“high-order” interactions (e.g., Levine et al. 2017) and differences in the diversity as
well as the identity of species across environmental gradients. This complexity is part
of the reason why we believe it has been difficult to clearly interpret how competition
contributes to setting range limits across environmental gradients when these factors
(diversity and identity) are not controlled, as they are in our experiment. We
acknowledge that focusing on pairwise competition is a simplification, but our
experimental design allows us to separate out effects of competitor identity from
changing intensity of interactions among particular species, and furthermore to gain
insight into how changing competition intensity is mediated by niche and fitness
differences between species. This insight would be much more difficult to obtain from
investigations within multispecies communities. We acknowledge that the net effect
of all interactions in natural systems might lead to quantitatively different conclusions
about where the range limits of our study species are set in nature, and have added
this caveat to the revised Discussion (lines 204-206 and 274-277). We also make
suggestions for future work to examine how complex interaction networks can

contribute to setting range limits (lines 317-320). Please also see our response to the
comment of reviewer 3 and 4 below regarding predictions of coexistence among
cooccurring species.

2. Perhaps the most interesting result in this paper is that niche differences tend to
modulate coexistence among sympatric but not parapatric species. However, I found
this result somewhat unsatisfying in that the biological mechanism driving this result
is not at all clear. While the authors do identify one mechanism that might be driving
this effect (phenology), it is not clear to me from their explanation whether such a
mechanism might be universal or not, particularly since they provide no biological/
physiological explanation for why phenology overlaps more outside of a species'
range rather than within the range.

We agree with the reviewer that the quantities of niche differences per se tell us little
about biological mechanisms. Their phenomenological nature is, in fact, a strength,
because they allow us to detect the operation of niche processes without a priori
hypotheses about the precise biological mechanisms, for which specific and targeted
experiments (e.g., manipulating nutrient concentrations, natural enemy pressure,
etc.) are needed. This is outside of the scope of the current paper, but we have
added this point to the future perspectives in the revised manuscript (lines 311-317).

Phenology has been shown to be a key trait modulating niche differences and
species coexistence in other systems (e.g., Usinowicz et al. 2017), but we, of course,
cannot conclude from our study about whether species in general have smaller
phenological overlap within versus beyond their range. We have explicitly included
this point in the revised manuscript (lines 223-228). In sum, we agree with the
reviewer that it would be satisfying to understand the mechanisms explaining
changes in niche overlap in further detail, but to do so requires a different set of
experiments that are beyond the phenomenological framework we adopt in the
current study.

3. Somewhat related to no. 2, I think the explanation of the Carroll et al. approach
needs quite a bit more explanation earlier in the paper, ideally with a biological
example to illustrate what these values actually reflect—perhaps even with a
conceptual figure of some kind. It is difficult to intuit what the ND v. RFD values
actually mean biologically, even after carefully reading the methods a number of
137 times and looking at the Carroll paper. I think this problem is exacerbated by the
138 switching back and forth between “niche difference” v. “niche overlap” in the text.

We thank the reviewer for pointing this out. We agree with the reviewer that the ND
and RFD concepts are abstract since they are only phenomenological syntheses of
biological mechanisms underlying species coexistence. To facilitate the intuitive
understanding of ND and RFD, we have fleshed out the concepts of ND and RFD in
the Introduction and accompanied them with specific biological examples (lines 59-
64), explicitly linked them with range limits (lines 65-67 and 73-76) in the Introduction
and linked the concepts with their calculations in the Methods (lines 447-455). We
have updated our terminology, using “niche differences” consistently wherever
possible to avoid confusion.

Minor comments:

Line 14-15: I found this sentence much too vague. I suggest inserting “the
importance of” after “how”. I also found “processes” too vague.

We have rephrased this sentence.

Line 48: change “species interactions”
to “competition”, as this paper is only focusing on competition, not on all species
interactions

We have updated using competition wherever suitable throughout the manuscript.

Line 124: Missing “growth rates” after ‘intrinsic population.’

We have rephrased this.

The fact that lowland species are predicted to occur at the highland site (but
currently do not) is troubling. It is unclear if this phenomenon arises from disconnect
between modelling results and reality, or whether it is actually due to disequilibrium
between current climate and the species’ distributions. The authors’ claim that it is
the latter would be strengthened by some evidence of dispersal limitation.

This is a good point. We have now explicitly discussed the possible roles of dispersal
limitation in giving rise to disequilibrium in lowland species’ distribution and added
relevant references (lines 264-266). This comment also made us think about other
possible reasons why some lowland plants might have performed better than
expected at high elevation in our experiment. One possibility is that while average
climatic conditions at high elevation (captured by our experiment) might be
permissive for these species, climate extremes might not. Extreme events (such as
late frosts) might contribute to upper range limits of lowland species but occur
infrequently, and so be missed by short-term experiments. We now include a
discussion of this possibility in our revision (lines 256-258).

What is the AIC (or AICc?) weight of the top models?

We used AICc and added the AICc weight in Supplementary Table 6 and another
table including the complete comparison of all candidate models in Supplementary
Data 1.

It appears that the issue of size eviction from the IPMs was assessed visually
somehow? I’m not sure how you would do this—a more methodological approach
(i.e., testing for any eviction) would be more appropriate.

Thank you for this suggestion. In the revised manuscript we now follow the method
suggested by Ellner et al. 2016 (pages 45-48) to detect the size eviction, and report
the results of this analysis in the revised Supplementary Methods that shows the size
eviction occurs with only a small probability (Supplementary Information lines 85-87).

I do not understand the rationale behind randomly selecting 56 IPMs in the
supplemental information. More clarification is needed.

We are sorry that the original statement was not clear. We integrated IPMs using
mid-point rules in which the mesh points (size range divided by the number of bins)
should be small enough to include all possible sizes of offspring. In other words, the
more bins the IPMs have, the more accurate the projected population growth is, but
the longer it takes to compute. To find the minimum number of bins on which
population growth rates converge, we projected the IPMs starting with 100 bins and
increased it until the projected population growth rates stabilized, an approach
suggested by Ellner et al. (pages 48-49) . We have fleshed out this procedure in the
Supplementary Methods (lines 87-89) and excluded the original figure including the
56 IPMs that we don't believe is necessary to include in the revised manuscript.

In the methods the authors indicate that the monoculture gardens weren't fully
established until autumn 2017, but then they use data from the 2017-2018 transition
to parameterize their IPMs—so this transition began before the monocultures were
fully established and should be excluded.

The reviewer is correct that not all monocultures were fully established in autumn
2017. To ensure that focal plants only compete against established monocultures,
we did not transplant any plants into those plots in autumn 2017. Therefore, the data
included in the analysis between 2017 and 2018 were measured only on focal plants
that competed against established monocultures. We realized the original statement
was unclear and clarified this statement in the revised Methods (lines 375-376).

Reviewer #3 (Remarks to the Author):

This paper combines field experiments with model simulations to look at species
range limits using modern coexistence theory. I like the design of this study that
directly tests the effects of intra- and interspecific competition and abiotic factors on
species coexistence along the elevation. I am impressed by the detailed fieldwork
conducted by the authors.

Thank you for your suggestions, particularly those on the modelling and analysis.

However, I have a few suggestions regarding the authors' analysis and presentation
of results.1. As the authors correctly understand, the resident populations of
competing species must be at equilibrium when conducting invasion analyses. That
is, the population growth rate of the competing species in interspecific competition
treatment should be close to zero. However, I did not see the authors address this
issue. From Figure S5, it appears that the population growth rates of intra-specific
competition treatment are much greater or less than zero in many species. This
means that populations are either declining or increasing. Is this also true for
resident species in the inter-specific competition treatment? Furthermore, although it
is reasonable that the population growth rate for the intra-specific competition is
lower than that for interspecific competition (because intra-species competition is
higher than the interspecies competition), the population growth of focal species is
usually very high, much higher than that of intra-species competition treatment, as
shown in Figure S5. This part is very strange to me, do the authors have any
explanation? This also leads me to ask whether the competing populations have not
reached equilibrium. I hope this part can be discussed more clearly because it is a
key assumption of the modern coexistence theory.

There are two issues here. Firstly, the reviewer correctly says that the “the resident populations of competing species must be at equilibrium when conducting invasion analyses”. That is, the monoculture plots should represent a population near its single-species carrying capacity, into which focal species are invading. We address this point in the next paragraph, below. Secondly, the reviewer says “the population growth rate of the competing species in interspecific competition treatment should be close to zero”, and is surprised that the growth rates for interspecific competition is often much higher than that of intraspecific competition. We believe this is a misunderstanding – in our experiment we measure *invasion* or *low-density* growth rates, that is, the population growth rate when the focal species is experiencing no conspecific density dependence (see lines 405-410). If interspecific competition is very weak (e.g., niche differences are very large), then invasion growth rates could be similar to intrinsic growth rates (that is, growth rates in absence of any competition). Therefore, we do not assume that interspecific competition has reached equilibrium, rather the opposite. We have included a diagram to clarify the design of our field experiment in Supplementary Fig. 1.

We further explored our data to address the reviewer’s question about whether our monoculture plots could be considered to be close to equilibrium. In the revised manuscript, we determined the uncertainty around our estimates of population growth rates using parametric bootstraps as suggested by the reviewer below. We took advantage of these bootstraps to explore whether the background monocultures were at equilibrium. Specifically, the 95% confidence interval of 32% (11 of 34) of intraspecific invasion growth rates (y-axis, log-transformed) included zero, indicating that these monocultures did not significantly depart from equilibrium (Supplementary Fig. 5). For the remaining monocultures that were predicted to depart from equilibrium, ten were predicted to be above equilibrium abundance ($\ln(\text{intraspecific invasion growth rates}) < 0$) and 13 below equilibrium abundance ($\ln(\text{intraspecific invasion growth rates}) > 0$); these case were evenly distributed across the sites (Supplementary Fig. 5), and therefore will not have biased our findings. We are grateful to the reviewer for raising this important point, and now discuss these considerations in the revised Discussion (lines 290 - 295).

2. This also makes me want to see more original data points in the authors' results. The authors provide the formula used to estimate each fitness component in Table S5, as well as the AIC values for the model described in the subsequent table. However, in the absence of comparisons, the AIC values do not provide much information. Therefore, I would like to see direct plots with actual data points of the relationship between the data points and the regressions. I would also like to see the regression plots with and without outliers (authors said they remove less than 1% of the data points) and the effect of these outliers on the results.

The original statement in the Reporting Summary was not clear. We excluded outliers when we fit the regression models that were used to estimate plant size and fecundity (see Data exclusion in updated Reporting Summary), only if obvious errors were identified (e.g., biologically unrealistic size or stalk height). We included already a figure to show the estimated plant sizes against the actual sizes in Supplementary Fig. 3, which allows readers to assess the performance of size regression models.

As requested, we have now included an additional figure to show the fitted vital rates
implemented in the models against the raw data in Supplementary Fig. 4. In addition,
we have added a complete comparison of all candidate models and the AICc weight,
as suggested by reviewer 2, in Supplementary Data 1 and Supplementary Table 6.

3. More formally, as the authors cite in Ellner's (2016) book on IPM estimation (p.
30), Ellner suggests that "Always quantify your uncertainty!", I would like to see the
authors estimate the uncertainty of the results and then separate the conclusions for
species with higher uncertainty from those with higher statistical power.

Thank you for this suggestion. In the revised manuscript, we performed parametric
bootstraps on size-dependent vital rates (i.e., survival, growth, flowering, and
fecundity) to account for uncertainty in our results. We resampled the parameters of
each vital rate 500 times from multivariate normal distributions using their means and
covariance matrices (lines 427-432). To account for the uncertainty around the
estimates of population growth rates, and to account for error propagation through
subsequent, we fitted all IPMs, calculated λ and quantified competitive outcomes and
coexistence metrics using the 500 bootstrap replicates (lines 471-472). Instead of
separating the conclusion for species with high vs low uncertainty (the separation
criteria would be arbitrary), we performed individual tests for each bootstrap replicate
and determined the significance of effects based on whether the 95% confidence
interval of a given effect included zero (lines 499-504). We have updated the Results
section with the bootstrapped results and included uncertainty estimates in all figures
in the revised manuscript (see Results and figures). Our original conclusions remain
after accounting for uncertainty in estimates of population growth rates.

4. I read the author's description of the experimental methods many times before I
understood how the authors' setups of the intra- and interspecific competition
treatment. I suggest that the authors could draw a diagram so that readers can
understand how intraspecific competition is created and how the positions of the
resident and invasive populations are arranged in the interspecific competition
treatments.

Thank you for the suggestion. We have added a diagram of the field experiment
design in Supplementary Fig. 1.

Reviewer #4 (Remarks to the Author):

It was a delight to read this manuscript that uses decent macroecological
experiments to examine how biotic interactions shape species distribution
boundaries. Such studies are relatively rare, but they are crucial in clarifying the role
of complex species interactions. Particularly, the authors applied coexistence theory
to understand the processes of competition underlying boundary formation. I had
many questions at the beginning of reading this manuscript, but most of these were
resolved as I continued. Overall, this is an important study with rigorous experiments,
and the manuscript is well-structured.

Thank you. We are delighted that the reviewer considers our work important and
rigorously conducted.

I have a few comments for the authors. Please forgive me for being a bit picky about
the concluding sentences in the abstract. “Taken together, these results challenge
the view that competition has a diminished role in structuring communities in
abiotically stressful environments.” This sentence insinuates a comparison between
competition and climate factors, but testing their relative importance was outside the
main focus of this study. The experiments were more about 'how' competition
processes affected distribution across environmental gradients. The setting of
experiments and analytical strategy will inevitably underestimate the effects of abiotic
factors, including the use of the same soil substrate, the short time frame for
parameter collection, which does not truly reflect how environmental fluctuation
affects population growth, and largely focused on niche difference versus fitness
differences. In any case, environmental effects are inferential and relative
contributions to competition are difficult to present. The authors are well aware of
these limitations in the discussion. I suggest that the abstract could be modified to
explicitly reflect the focus of the study.

This is a good point. We agree with the reviewer that our original narrative, including
“abiotically stressful environments”, might have distracted readers from our focus on
the role of competition in shaping species distributions. We have reframed the
abstract to focus on competition. We have also rephrased the sentence mentioned
by the reviewer (lines 296-299), although we maintain that our results do implicate
competition as an important factor affecting range limits at high elevation.

I also suggest that the Introduction paragraph can be presented in a hypothesis-
prediction format. The application of coexistence theory in biogeographic studies at a
large scale and range limits is still in its infancy (e.g. Alexander et al. 2018 TREE). Its
novelty and importance are noted, but not necessary to readers outside the field.
The authors could provide more context to smoothly bridge these fields. Particularly
in the context of environmental gradients, the so-called harsh or benign climate, what
are the expected consequences for niche difference and relative fitness, based on
known hypotheses.

We thank the reviewer for the suggestion. In the revised Introduction we have now
elaborated on the concepts of niche and fitness differences, providing predictions
and examples for how these might change across an elevation gradient (lines 59-64)
and emphasized the novelty of the link between coexistence theory and species'
range limits more explicitly (lines 65-69 and 73-76).

The population models are essential to derive the coexistence metrics, niche
differences, and relative fitness differences. However, because the parameters were
obtained at a short time period, which does not reflect how environmental fluctuation
affects species coexistence, the metrics may be systematically biased, probably
leading to the underestimation of species coexistence at high altitudes. This may be
beyond the scope of this study, but the authors can discuss the possible effects.

Thank you for this suggestion. We have explicitly discussed these limitations of our
study in the revised manuscript (lines 256-258 and 279-290).

The results showed that neither competition nor climate limits lowland species, and
their current distribution reflect disequilibrium with climates. The argument will then

suggest that the current experimental setting may be somewhat biased and the
results can be misleading to a certain extent.

We take the reviewer's point but don't believe that our results would be biased, even
if current distributions are in disequilibrium with climate. The results in Fig. 1 suggest
that if we were to increase the elevation of the high site, we would still expect the
range limits of lowland species to occur firstly in the presence vs. absence of
neighbors. This suggests that our conclusion that competition is important for setting
range limits at low and high elevations remain valid. We now add this point to the
Discussion (lines 269 – 271).

The authors use a community perspective to analyze how the upper and lower
boundaries of species are affected by competition, but the comparisons are between
the upper boundaries of lowland species and the lower boundaries of highland
species, which are all adjacent to other species, not at the harsh boundary of the
target gradient, and not comparing the upper and lower boundaries of specific
species. As the stress gradient hypothesis can be largely from a species
perspective, discussing the relative pressure at their upper and lower boundaries, it
is suggested that the authors clarify the differences.

The reviewer is right that our study sites were located beyond the upper boundary of
lowland species (the high site) or beyond the lower boundary of highland species
(the low site). To avoid any confusion about this aspect of the design, we now
explicitly refer to the upper boundary of lowland species and the lower boundary of
highland species throughout the manuscript (lines 91-95). We suggest that future
studies look at whether competition set limits across the whole range (lines 317-
320).

429 References

Ellner, S. P., D. Z. Childs and M. Rees (2016). Data-Driven Modelling of Structured
Populations : A Practical Guide to the Integral Projection Model, Cham : Springer.

Levine, J. M., J. Bascompte, P. B. Adler and S. Allesina (2017). "Beyond pairwise
mechanisms of species coexistence in complex communities." Nature **546**: 56.

Usinowicz, J., C.-H. Chang-Yang, Y.-Y. Chen, J. S. Clark, C. Fletcher, N. C. Garwood, Z.
Hao, J. Johnstone, Y. Lin, M. R. Metz, T. Masaki, T. Nakashizuka, I. F. Sun, R. Valencia, Y.
Wang, J. K. Zimmerman, A. R. Ives and S. J. Wright (2017). "Temporal coexistence
mechanisms contribute to the latitudinal gradient in forest diversity." Nature **550**(7674): 105-
108.

REVIEWERS' COMMENTS:

Reviewer #2 (Remarks to the Author):

I found that the edits to this manuscript dramatically improve the clarity, and I commend the authors on their work. I have two very small comments:

Line 359: which “other plots” are the authors referring to?

Line 468: In the discussion the authors claim that competition is important in setting range limits at both high and low edges, with no pattern consistent with the stress-gradient hypothesis, but then in the methods (line 468), they say that facilitative interactions were discarded—making it difficult to understand whether there is any support at all for the stress gradient hypothesis. In lines 475-476, perhaps they are referring to an analysis that included facilitative interactions, in which case this would be less concerning.

Reviewer #3 (Remarks to the Author):

I am very glad that the authors have done so much analysis to address my questions. I think this paper is very good and one of the few clear experimental papers to test the species coexistence theory. I have no more questions and think this paper deserves to be published in Nature Communications!

Reviewer #4 (Remarks to the Author):

After reading the manuscript carefully, I think the authors have addressed my concerns. They better explained how niche and fitness differences may change across an elevation gradient and contexture by using coexistence theory to test the processes shaping species' range edges. They also explicitly discussed limitations, including climate fluctuation and above-below ground interaction, which leads to necessary future research. They have improved the statistics and treated the evidence appropriately to generate their discussion and conclusion. I am satisfied with the current revision.

**Author responses**

Reviewer #2 (Remarks to the Author):

I found that the edits to this manuscript dramatically improve the clarity, and I commend the
authors on their work. I have two very small comments:

Thank you very much for your comments!

Line 359: which “other plots” are the authors referring to?

Thanks for pointing this out. “other plots” refers to the plots that failed to establish in autumn
2017. We have added “the other plots that failed to establish” for clarity (lines 356-357).

Line 468: In the discussion the authors claim that competition is important in setting range
limits at both high and low edges, with no pattern consistent with the stress-gradient
hypothesis, but then in the methods (line 468), they say that facilitative interactions were
discarded—making it difficult to understand whether there is any support at all for the stress
gradient hypothesis. In lines 475-476, perhaps they are referring to an analysis that included
facilitative interactions, in which case this would be less concerning.

Thank you for pointing this out. We only excluded facilitative interactions to calculate niche
and fitness differences because this analysis is not possible with positive interactions.

Therefore, our main result related to competition across the gradient (Fig. 1) and the
analysis you referred to in lines 475-476 (now removed) did include the facilitative
interactions. We have added “we did not exclude facilitative interactions for other analyses”
to the corresponding section in Methods for clarity (lines 470-471).

Reviewer #3 (Remarks to the Author):

I am very glad that the authors have done so much analysis to address my questions. I think
this paper is very good and one of the few clear experimental papers to test the species
coexistence theory. I have no more questions and think this paper deserves to be published
in Nature Communications!

Thank you very much for your comments!

Reviewer #4 (Remarks to the Author):

After reading the manuscript carefully, I think the authors have addressed my concerns.
They better explained how niche and fitness differences may change across an elevation
gradient and contexture by using coexistence theory to test the processes shaping species'
range edges. They also explicitly discussed limitations, including climate fluctuation and
above-below ground interaction, which leads to necessary future research. They have
improved the statistics and treated the evidence appropriately to generate their discussion
and conclusion. I am satisfied with the current revision.

Thank you very much for your comments!
